# Differences in Associations of Three Types of Alcoholic Beverages with Age-Related Cognitive Decline in Men

**DOI:** 10.3390/nu16213714

**Published:** 2024-10-30

**Authors:** Marie Stjerne Grønkjær, Trine Flensborg-Madsen, Merete Osler, Holger Jelling Sørensen, Ulrik Becker, Erik Lykke Mortensen

**Affiliations:** 1Center for Clinical Research and Prevention, Copenhagen University Hospital—Bispebjerg and Frederiksberg, 2000 Frederiksberg, Denmark; merete.osler@regionh.dk; 2Department of Public Health, University of Copenhagen, 1353 Copenhagen, Denmark; tfl@sdu.dk (T.F.-M.); elme@sund.ku.dk (E.L.M.); 3Center for Healthy Aging, University of Copenhagen, 2200 Copenhagen, Denmark; 4National Institute of Public Health, University of Southern Denmark, 1455 Copenhagen, Denmark; holgerjsorensen@gmail.com (H.J.S.); ulbe@sdu.dk (U.B.)

**Keywords:** alcohol consumption, alcoholic beverages, wine, beer, spirits, cognitive decline, health effects, population-based studies, adults, aging

## Abstract

Objectives: To investigate the influence of wine, beer, and spirits consumption, respectively, on non-pathological, age-related cognitive decline from young adulthood to late midlife in a large follow-up study of Danish men. Methods: The study includes 2456 middle-aged Danish men from the Lifestyle and Cognition Follow-up study 2015, with information on adult-life consumption (from age 26) of wine, beer, and spirits self-reported in late midlife and age-related cognitive decline assessed using the same validated intelligence test administered in young adulthood and late midlife. Associations were adjusted for consumption of other alcoholic beverages, year of birth, age at follow-up, retest interval, education, young adulthood intelligence, and personality. Results: Most of the men had wine (48%) or beer (42%) as their preferred beverage type. For all three alcoholic beverages, consumption of more than 14 units weekly was associated with a greater decline in unadjusted analyses, but this trend was only significant for wine. In contrast, adjusted models showed that moderate wine and spirits consumption was associated with less decline than abstention for these alcohol types (*p* = 0.03 for 8–14 units/week of wine and *p* = 0.03 for 1–7 units/week of spirits). Statistical tests suggested a difference between the estimated effects of consumption of 8–14 units/week of wine and beer on cognitive decline. Conclusions: While patterns of associations were similar across beverages, moderate wine and spirits consumption may mitigate cognitive decline, in contrast with beer. However, the results should be interpreted with caution due to inherent differences between men with different alcoholic beverage preferences.

## 1. Introduction

Cohort studies of the association between alcohol consumption and non-pathological, age-related cognitive decline have suggested larger cognitive decline with both alcohol abstinence [1,2] and heavy alcohol consumption [3,4,5,6] compared with moderate alcohol consumption. These associations are, to some extent, similar to the associations observed between alcohol consumption and other health outcomes, such as ischemic heart disease [7], dementia [8], and mortality [9].

The observed associations between alcohol consumption and age-related cognitive decline may be caused by alcohol per se, other compounds in alcoholic beverages, or confounding factors, such as social status or smoking. If it is alcohol per se that influences age-related cognitive decline, similar associations would be expected across different alcoholic beverages. A recent meta-analysis based on 12 studies concluded that wine consumption may be associated with lower risk of dementia and cognitive impairment [10]. However, only a few studies have investigated whether associations with non-pathological, age-related cognitive decline (i.e., including at least two cognitive assessments) vary between different types of alcoholic beverages [2,11]. In the study by Nooyens et al. [11], the amount of red wine was inversely associated with a decline in cognitive ability over a 5-year period, whereas none of the other types of alcoholic beverages were associated with cognitive decline. The study by Stampfer et al. [2] found that moderate consumption of wine as well as other types of alcoholic beverages was associated with a lower risk of substantial decline in cognitive performance over a 2-year period in 70–81-year-old women. Both studies concluded that they did not include enough heavy drinkers in the study sample to assess the effect of high levels of alcohol intake. Moreover, the short retest intervals used to assess cognitive changes in both studies and the high age at baseline in the study by Stampfer et al. [2] make it difficult to identify (small) group differences in changes over time in cognitive ability. Thus, studies are needed that investigate the association between different types of alcoholic beverages and age-related cognitive decline using study samples that include enough individuals with heavy alcohol consumption and sufficiently long retest intervals to assess age-related cognitive changes. Identification of risk factors, such as different types of alcoholic beverages, for age-related cognitive decline is important for public health, as it can inform guidelines and risk assessments, thus ultimately contributing to better prevention strategies and healthier aging [12].

The aim of the present study was to investigate the influence of wine, beer, and spirits consumption, respectively, on non-pathological, age-related cognitive decline from young adulthood to late midlife in a large follow-up study of Danish men. The data used are unique for this purpose, as they include a reliable assessment of age-related cognitive decline (using the same intelligence test with a mean retest interval of 41 years), detailed information on alcohol consumption, a rather large number of heavy drinkers, and information on important covariates.

## 2. Materials and Methods

### 2.1. Study Sample

The Lifestyle and Cognition Follow-up study 2015 (LiKO-15) consists of 2611 men who completed the same global intelligence test in young adulthood (mean age of 20 years) and in late midlife (mean age of 62 years) [3]. The men first completed the intelligence test as a paper-and-pencil test as part of the military draft board examination between 1968 and 1989, whereas they were retested using a computerized version of the same test in the LiKO-15 study between 2015 and 2017. The military draft board examination is mandatory; hence, only men with certified disqualifying diseases are exempted, corresponding to approximately 5–15% [13]. The men invited to the LiKO-15 study were selected among men who had been draft-board-examined in Greater Copenhagen or the island of Bornholm and were currently living within 50 km of test locations [3]. The overall aim of the LiKO-15 study was to investigate alcohol and cognition, and a large proportion of men with psychiatric hospital diagnoses, including alcohol use disorders, were invited to participate. A total of 19,888 men selected from a previous database on IQ and mental disorders and from the Danish Conscription Database were invited to participate in the LiKO-15 study, and 2611 men agreed to participate in the study (13% participation rate) [14]. Unfortunately, data on 89 participants were lost due to technical problems with the military’s computer systems, and, due to an error in the setup of the questionnaire at the beginning of data collection, 42 participants were not asked questions about personality. Except for these 131 men, most participants in LiKO-15 provided complete information for all included items in the questionnaire, and, for the present paper, 2456 men with complete information on intelligence and all variables of interest were included in the analyses.

### 2.2. Variables

#### 2.2.1. Age-Related Cognitive Decline

Changes in IQ scores from young adulthood to late midlife were used to assess age-related cognitive decline. The intelligence test that participants completed at the draft board examination and at the follow-up examination was Børge Priens Prøve (BPP), which is a 45 min global test of fluid intelligence. The test was administered by specially trained military staff at the draft board examination, and such staff also administered the test at the midlife follow-up. The test consists of a total of 78 items within subtests of letter matrices, verbal analogies, number series, and geometric figures [13]. The total BPP score has been shown to be highly correlated (r = 0.82) with the score on the Wechsler Adult Intelligence Scale (WAIS). The total score on BPP ranges from 0 to 78 points, and both sets of raw test scores (from young adulthood and late midlife, respectively) were linearly standardized to an IQ scale with a study sample mean of 100 and a standard deviation of 15 in young adulthood. The IQ scale is generally known and frequently used, thus making our results easier to interpret and compare to results in other studies.

#### 2.2.2. Alcoholic Beverages

Participants completed a comprehensive questionnaire at the follow-up examination in late midlife, including detailed information on lifetime alcohol consumption split into beer, wine, and spirits. The alcohol consumption questionnaire is described in detail elsewhere [3]. For the present paper, the primary alcohol consumption variables were constructed using self-reported retrospective information on the following question: ‘How many units of alcohol did you typically drink a week split on beer, wine, and spirits?’ The question was split into the following age periods: 26–30 years, 31–40 years, 41–50 years, 51–60 years, and 61 years or older. Information was available for younger age periods (14 years or younger, 15–18 years, and 19–25 years), but our focus was on adult-life consumption, i.e., consumption during the period between the two cognitive assessments. In Denmark and in our study, one unit of alcohol is defined as 12 g of pure alcohol. The phrase ‘a unit [of alcohol]’ is commonly employed by the general population, and it is widely understood that one Danish unit of alcohol roughly corresponds to a 33 cl beer or a standard glass of wine. The mean weekly adult-life alcohol consumption (total and split into wine, beer, and spirits) was calculated as average units of alcohol consumed per week and averaged across age periods. *The preferred alcoholic beverage consumed throughout the individual’s adult life* was based on the percentage of the total weekly adult-life consumption of wine, beer, and spirits, respectively. Based on these percentages, the men were divided into the following categories: no alcohol consumption; wine preference [highest percentage of wine consumption]; beer preference [highest percentage of beer consumption]; spirits preference [highest percentage of spirits consumption]; and no clear preference [men with identical percentages of two or more types of alcohol]. *The adult-life weekly units of wine, beer, and spirits* were divided into the following consumption groups: 0; 1–7; 8–14; and ≥15. These groups were selected because they are divisible by seven, thus allowing for comparison with studies that report results in units per day. In a previous study on total alcohol consumption and cognitive decline using the same study sample [3], we found that consumption of 28 units of alcohol per week or more was associated with a larger cognitive decline. Nevertheless, to avoid too few participants in the highest consumption group when data were stratified based on the different types of alcoholic beverages, consumption of 15 units or more was used as the highest consumption group in the present study.

#### 2.2.3. Covariates

Information on the *year of birth* was obtained from the Danish Civil Registration System, and *age at follow-up* was calculated using information regarding the date of participation in the follow-up examination, whereas the *retest interval length* was calculated using additional information on the date of participation in the draft board examination. *Young adulthood IQ* based on Børge Priens Prøve completed at the draft board examination was also included as a covariate. In addition, *years of education* were calculated based on the nominal study length and self-reported information on school and vocational education obtained from the questionnaire completed in late midlife. Finally, the Danish version of the 10-item short version of the Big Five Inventory (BFI-10) completed by participants at the follow-up examination was used to assess *personality traits* [15]. BFI-10 includes two items for each of the five personality traits—extraversion, agreeableness, conscientiousness, neuroticism, and openness—scored from 1 to 5 (disagree strongly–agree strongly). The average of the two items for each trait (with one of them reversed) was calculated. Previous studies have shown that personality may influence both alcohol consumption and cognitive aging. Although personality was only assessed at the follow-up examination, it was included as a covariate because personality traits are assumed to be relatively stable during adult life.

#### 2.2.4. Additional Variables

Additional variables related to alcohol consumption patterns, other health behaviors, and morbidity were also included; however, these variables were not included in the primary analyses because they are not necessarily confounders, as they can also mediate associations.

Participants were asked at the follow-up examination ‘How often have you typically consumed 10 units [of alcohol] or more on the same occasion in the different age periods?’ Based on self-reported frequency (seven response categories from never/almost never to every day/almost every day) for each adult-life age period (26–30 years, 31–40 years, 41–50 years, 51–60 years, and ≥61 years) as well as the duration of the period, the *number of years with weekly extreme binge drinking* throughout the individual’s adult life was calculated.

Information on other health behaviors, including current (physical activity) or lifetime (smoking and psychoactive drug use) behaviors, was also self-reported at the follow-up examination. *Pack-years of smoking* was calculated based on the following tobacco equivalents: 1 cigarette = 1 unit, 1 cigarillo = 3 units, 1 cigar = 5 units, and 1 pipe = 3 units [16]. *Physical inactivity in leisure time* included participants who reported that they were primarily sedentary in their leisure time. Finally, participants were asked at the follow-up examination ‘How often have you used psychoactive drugs in the different age periods?’ Based on self-reported frequency (seven response categories from never/almost never to every day/almost every day) for each adult-life age period (26–30 years, 31–40 years, 41–50 years, 51–60 years, and ≥61 years) as well as the duration of the period, the *number of years with weekly use of psychoactive drugs* throughout the individual’s adult life was calculated.

Information on alcohol-related disorders and psychiatric and somatic morbidity was obtained by linking study participants to the Danish Psychiatric Central Research Register [17] and the Danish National Patient Registry [18] using the unique personal identification number assigned to all Danish residents [19]. The following diagnostic codes were used to classify *alcohol-related hospital diagnoses*: 291.09–291.99, 303.09–303.99, 571.09, and 571.10 in ICD-8 and F10.1–F10.9, E24.4, G31.2, G62.1, G72.1, I42.6, K29.2, K70, K85.2, and K86.0 in ICD-10. To classify psychiatric morbidity in terms of *other mental disorder hospital diagnoses* from psychiatric departments, the following diagnostic codes were used: 290–304 in ICD-8 and F10–99 in ICD-10 (e.g., including mental and behavioral disorders due to psychoactive substance use; schizophrenia, schizotypal, and delusional disorders; affective disorders; and neurotic, stress-related, and somatoform disorders). The *Charlson Comorbidity Index* [20] was calculated to assess the burden of somatic morbidity.

### 2.3. Statistical Analyses

The overall characteristics of the study sample are presented in a previous paper that used the same study sample to investigate total adult-life alcohol consumption and cognitive decline [3]. Characteristics of the study sample categorized by preferred alcoholic beverage throughout adult life were examined, and differences between groups were tested using the chi-squared test for categorical variables and one-way ANOVA for continuous variables. Associations between adult-life wine, beer, and spirits consumption, respectively, and IQ changes were examined in unadjusted and adjusted linear regression analyses. Two adjusted models were included: 1. including adjustment for consumption of other alcoholic beverages (i.e., mutual adjustment by including weekly units of wine, beer, and spirits in the same model) (Model 1) and 2. further adjustment for year of birth, age at follow-up, retest interval length, young adulthood IQ, years of education, and personality (Model 2). In all models, consumption levels of each type of beverage were compared to the group who abstained from this type of beverage (0 units per week).

Supplementary analyses were conducted by adjusting for additional variables, including extreme binge drinking throughout adult life, other health behaviors, and morbidity.

To further evaluate differences between the estimated effects of alcoholic beverage types on cognitive decline, the significance of differences between regression coefficients for wine, beer, and spirits was evaluated at each consumption level (1–7 units, 8–14 units, and ≥15 units). Overall tests of the three types of beverages can be conducted, but to obtain detailed information on any differences, pairwise comparisons of coefficients were conducted (9 comparisons for each statistical model).

All analyses were carried out using Stata versions 15 and 18 (StataCorp LLC, College Station, TX, USA).

## 3. Results

### 3.1. Preferred Alcoholic Beverage of the Study Sample

As described in a previous paper, men included in the study sample had an average decline of 5.1 IQ points from young adulthood (mean age = 20.3 years, SD = 2.1) to late midlife (mean age = 61.6 years, SD = 3.3), an adult-life average alcohol consumption of 12.4 (SD = 13.1) units per week, and 3.9 (SD = 9.5) years with weekly extreme binge drinking [3]. Most of the participants had wine as their preferred type of alcoholic beverage (48.4%), followed by beer (42.4%), whereas only 2.3% of participants preferred spirits, 2.8% did not consume alcohol throughout their adult life, and 4.2% did not have a clear alcoholic beverage preference (Table 1). The highest mean IQ score in young adulthood and late midlife as well as the highest number of years of education were observed among men who preferred wine. Only small differences in personality traits were observed across the preference groups, including lowest scores on conscientiousness in men with a spirits preference and highest scores on neuroticism in men with no clear preference. The highest adult-life alcohol consumption—in terms of both average weekly consumption and weekly extreme binge drinking—was observed among men who preferred spirits, followed by men preferring beer and men preferring wine. A similar pattern—i.e., highest prevalence in men with a spirits preference and lowest prevalence in men with a wine preference—was observed in relation to other unhealthy behaviors, such as smoking, and morbidity, including, for example, alcohol-related hospital diagnoses. More than a third of the men who preferred spirits had such diagnoses.

### 3.2. Wine, Beer, and Spirits Consumption and Cognitive Decline

In the unadjusted analyses, moderate consumption (1–7 units) was associated with a similar IQ decline as no consumption (0 units per week), whereas high consumption (≥15 units/week) was associated with a larger IQ decline than no consumption (Table 2), although this association was only significant for wine (*p* = 0.031). The overall pattern was similar for all three beverage types, and statistical tests showed no differences between regression coefficients at each consumption level. For analyses including mutual adjustment for consumption of other alcoholic beverages (Model 1), a similar pattern was observed, with a 2.4 IQ points larger IQ decline in men with high consumption of wine (*p* = 0.015) than in men with no consumption of wine, but the comparison of the corresponding regression coefficients for the three types of beverages showed no differences.

Further adjustment for sociodemographic factors and personality (Model 2) substantially changed the results for wine and spirits. Weekly consumption of 1–7 units of wine was associated with a marginally significant smaller decline, while consumption of 8–14 units of wine was associated with a decline that was 1.5 IQ points lower (*p* = 0.030) than no consumption of wine. For spirits, weekly consumption of 1–7 units was associated with a decline that was 0.8 IQ points lower (*p* = 0.030), while consumption of 8–14 units was associated with a decline that was 1.3 IQ points lower. The latter estimate was not significant, which probably reflects a lack of statistical power. The test of differences between corresponding regression coefficients showed that the wine coefficient for 8–14 units (Β = 1.47; *p* = 0.030) was different from the corresponding coefficient for beer (Β = −0.76; *p* = 0.321). In general, the results for beer were similar for the unadjusted and the two adjusted statistical models.

### 3.3. Supplementary Analyses

These analyses included, in addition to the covariates in Model 2, further adjustments for the number of years with weekly extreme binge drinking, other health behaviors, and morbidity. Adjustments for extreme binge drinking and morbidity primarily attenuated the IQ decline associated with high (beer) consumption, whereas adjustments for other health behaviors showed results similar to Model 2 in Table 2 (Appendix A). The association of a reduced decline with moderate consumption of wine (8–14 units/week) and spirits (1–7 units/week) remained significant after adjustment for extreme binge drinking and other health behaviors, while only the association with moderate consumption of spirits was significant when adjusting for morbidity, including alcohol-related hospital diagnoses.

## 4. Discussion

### 4.1. Summary of Findings

In this large study sample of middle-aged men, the alcoholic beverage preference was primarily wine or beer. In unadjusted analyses and analyses mutually adjusted for consumption of other types of alcoholic beverages, weekly wine consumption of 15 units or more was associated with a larger IQ decline than no consumption of wine. Similar trends, although not statistically significant, toward larger decline with high weekly consumption were also observed for beer and spirits, and statistical tests showed no differences between the corresponding regression coefficients for the three types of beverages. However, when adjusting for potential confounders, the pattern changed for wine and spirits and showed associations between moderate consumption and less of a decline and a difference between the regression coefficients for consumption of 8–14 units of wine and beer.

### 4.2. Comparison with Previous Studies

Our findings of reduced cognitive decline with moderate consumption of wine and spirits are, to some extent, supportive of findings from previous studies. Nooyens et al. [11] found that low to moderate consumption of red wine was associated with less decline in global cognitive function over a 5-year period in a sample of 43–70-year-old men and women. In addition, Stampfer et al. [2], in a study of women (aged 70–81 years), found that the risk of substantial decline in cognitive performance over a 2-year period was significantly lower among moderate drinkers than non-drinkers, regardless of the type of alcoholic beverage consumed. These studies are not comparable to ours with respect to the age of participants and the follow-up interval, but a Scottish study with an assessment at age 70 was able to adjust for childhood intelligence and adult socioeconomic status [21]. When adjusted, most effects of alcohol disappeared, but small positive effects of wine and negative effects of beer remained significant. Similarly, we observed relatively small effects after adjustment for young adult intelligence and years of education. Thus, our findings extend previous findings and suggest that moderate consumption of wine and spirits during adult life (and not just current consumption) may be associated with less cognitive decline than abstention from these alcoholic beverages. A larger decline with high consumption of all three types of alcoholic beverages was also observed, but these estimates were not statistically significant after adjusting for potential confounders, which is possibly due to the relatively small number of participants with high weekly consumption of each type of alcohol.

Few studies have investigated the influence of alcoholic beverages other than wine on age-related cognitive decline [2,11], and our study may be the first to compare the three types of beverages using direct statistical tests of differences in corresponding regression coefficients. Thus, more studies are needed to confirm or refute our findings suggesting differences in the effects of moderate weekly consumption of wine and beer.

### 4.3. Interpretation of the Findings

Dissimilar associations with cognitive decline observed across different alcoholic beverages could indicate that it is not alcohol per se that influences cognitive decline. Thus, compounds other than ethanol in wine and spirits could explain the apparently beneficial effects of moderate consumption on cognitive decline. Primarily in relation to wine, there has been a focus on explaining the apparently beneficial effects on health, where polyphenols (besides ethanol) seem to be central in explaining red wine’s antioxidant, anti-inflammatory, and cytoprotective properties [10,22]. However, in our study, significant associations with less cognitive decline were only observed for wine and spirits after adjusting for potential confounders, and direct statistical tests of differences between corresponding regression coefficients for the three types of beverages were only significant for moderate consumption of wine and beer. Furthermore, this test was only significant at the 5% level, and this was in fact the case for all significant results in Table 2. Thus, none of these tests would remain significant if an adjustment was made for the relatively large number of statistical tests (e.g., Bonferroni correction).

For wine, and, to some extent, spirits, adjustment for potential confounders changed the associations from indicating a larger decline with high consumption to a pattern suggesting less of a decline with moderate consumption. A crucial factor may be adjustment for young adult intelligence, because, in this study sample, high initial IQ has been shown to be associated with larger cognitive decline [3], and, in our data, young adult IQ varies systematically from 93 (no wine consumption) to 105 (high consumption). Other factors may also have contributed to the effects of adjustment, as previous Danish studies have suggested that individuals who preferred wine had higher intelligence or cognitive ability [23,24] and were more advantaged in relation to all investigated social and personality factors [24]. The supplementary analyses adjusting for young adult IQ as well as extreme binge drinking, other health behaviors, and morbidity corroborated the associations between moderate consumption of wine and spirits and less decline. However, several unmeasured and unknown factors may confound estimates of the effects on cognitive decline and comparisons of the effects of the three beverage types. For example, dietary intake is associated with alcoholic beverage preference [25,26], but it is very difficult to measure accurately, and it can be questioned whether it is actually possible to adjust for all of the inherent differences between groups with different alcoholic beverage preferences.

Finally, some of our results may be related to subgroup sample sizes and associated statistical power. For example, the 15 units or more subgroups included 133 men for wine and 215 for beer but only 27 men for spirits. Thus, apparent differences in the effects of the three types of beverages on cognitive decline may, to some extent, reflect statistical power. The estimated coefficients in some cases corroborate this interpretation and suggest that even significant effects of alcohol consumption are small compared to the theoretical IQ standard deviation of 15 (adjusted for covariates, the largest effect was 1.47, which was associated with 8–14 units of wine, corresponding to 0.10 standard deviation). These positive effects are smaller than the negative effects on cognitive decline of more than 28 units a week observed in the same sample in a study of total alcohol consumption [3], and a particular weakness of the current study is that we were unable to analyze the effects of higher levels of consumption in detail. Thus, we cannot preclude that differences in the effects of the three beverage types could be demonstrated for higher consumption levels.

### 4.4. Strengths and Limitations

The present study has several strengths, including a relatively large sample size and high-quality assessments of alcoholic beverage consumption, age-related cognitive decline, and covariates. First, the detailed information on alcohol consumption enabled investigation of the influence of different alcoholic beverages on cognitive decline. Moreover, the detailed information on alcohol consumption throughout adult life made it possible to generate an adult-life (from age 26) abstinent group, thus reducing the risk of including previous heavy drinkers or individuals that are currently abstinent due to diseases in the abstinent group. Second, the quality of the assessment of age-related cognitive decline was also a clear advantage; the test is validated to assess global intelligence, it is sensitive to age-related changes, and it was administered with an exceptionally long mean follow-up interval of 41 years. Finally, we had detailed information on covariates, such as educational level, baseline IQ, pack-years of smoking, personality, and psychiatric and somatic morbidity.

The selection of participants in LiKO-15 is one of the main limitations of the present study. First, the generalizability of the results may be limited to Danish men without early-life diseases, because only men without certified disqualifying diseases were required to show up before the draft board. Second, the overrepresentation of men with psychiatric hospital diagnoses, including alcohol use disorders, ensured a relatively large proportion of men with heavy alcohol consumption in the study sample, but this could also introduce bias. Yet, we assume that this was not a major problem, as no statistically significant interactions between a binary psychiatric disorder variable and the exposure variables (wine, beer, and spirits consumption and wine/beer proportion) in relation to IQ decline were observed. Third, only 13% of those invited to participate in LiKO-15 decided to participate. Participation analyses published in a previous paper [3] revealed that this self-selection led to a study sample with a higher school educational level, higher baseline intelligence test scores, and a lower prevalence of mental disorders and other morbidities than non-participants. Unfortunately, we lack information on the reasons for non-participation, but, considering the approximately two-hour duration of the examination, the requirement to travel to a specific location for cognitive testing, and the lengthy questionnaire involved, it is plausible that these factors contributed to the rather low participation rate. Nevertheless, the selection has not necessarily biased the estimated cognitive decline within different levels of alcoholic beverages, as the study still had sufficient variance in all the included variables. However, our results may not be generalizable to women or countries with different drinking cultures.

The detailed information on alcohol consumption also has some limitations. The method used to obtain the information in terms of self-reported, retrospective recall of alcohol consumed in relation to different alcoholic beverages several years ago may limit the accuracy of the information. However, there is evidence that retrospective decade-based information on alcohol consumption is quite reliable [27], and our methods are only based on the assumption that the accuracy of the reported amount of alcohol consumed is sufficient to categorize participants into consumption groups. As such, we do not rely too much on the specific amount reported by each participant. Finally, the rather broad groups of alcoholic beverages included in the study may also have some limitations. For example, in addition to differences in compounds other than ethanol between alcoholic beverage types, there may also be differences within each type, e.g., between white wine and red wine or vodka and whiskey, which could influence the association with cognitive decline.

## 5. Conclusions

In a large study sample including men who primarily consumed wine and beer, the patterns of associations between consumption levels and cognitive decline were in general similar for the three types of alcohol (wine, beer, and spirits), although the results suggest that moderate consumption of beer may be associated with greater cognitive decline than moderate consumption of wine and spirits. These results should be interpreted with caution because of weak statistical power for some comparisons and because differences among alcoholic beverages in the associations with cognitive decline may not only reflect the effects of specific compounds but also inherent differences in both known and unknown characteristics between men who primarily consume wine, beer, or spirits.

## Figures and Tables

**Table 1 nutrients-16-03714-t001:** Characteristics of the male study sample grouped by preferred alcoholic beverage throughout adult life (N = 2456).

	Preferred Alcoholic Beverage Throughout Adult Life					
	No Alcohol Consumption	Wine Preference	Beer Preference	Spirits Preference	No Clear Preference	*p*-Value ^a^
**Total (N [%])**	68 (2.8)	1188 (48.4)	1041 (42.4)	56 (2.3)	103 (4.2)	
**Sociodemographic factors and personality**						
Year of birth (mean [SD])	1955 (3.1)	1954 (3.1)	1954 (3.3)	1955 (3.5)	1955 (3.4)	0.379
Age at draft board examination (mean [SD])	20.1 (1.9)	20.6 (2.2)	20.0 (1.8)	19.8 (1.6)	20.1 (2.0)	<0.001
Age at follow-up examination (mean [SD])	61.4 (3.1)	61.7 (3.2)	61.6 (3.3)	61.5 (3.5)	61.4 (3.5)	0.708
Retest interval (mean [SD])	41.3 (3.2)	41.2 (3.3)	41.6 (3.4)	41.7 (3.3)	41.3 (3.6)	0.049
Young adulthood IQ score (mean [SD])	95.7 (15.7)	103.3 (13.7)	96.8 (15.7)	101.1 (15.1)	96.0 (14.1)	<0.001
Late midlife IQ score (mean [SD])	91.1 (16.3)	97.9 (13.1)	91.9 (15.4)	95.3 (13.8)	91.2 (15.6)	<0.001
Changes in IQ score (mean [SD])	−4.6 (11.1)	−5.4 (8.9)	−4.9 (9.8)	−5.8 (8.1)	−4.8 (10.0)	0.668
Years of education (mean [SD])	13.1 (2.6)	14.3 (2.4)	12.8 (2.5)	13.3 (2.4)	13.2 (2.4)	<0.001
Personality traits ^c^						
Extraversion (mean [SD])	3.4 (1.0)	3.6 (0.9)	3.6 (0.9)	3.6 (0.8)	3.5 (1.0)	0.630
Agreeableness (mean [SD])	3.8 (0.8)	3.9 (0.7)	3.8 (0.7)	3.8 (0.7)	3.9 (0.7)	0.112
Conscientiousness (mean [SD])	4.0 (0.8)	4.1 (0.7)	3.9 (0.7)	3.6 (0.9)	4.0 (0.7)	<0.001
Neuroticism (mean [SD])	2.3 (0.9)	2.2 (0.8)	2.3 (0.9)	2.2 (0.8)	2.4 (0.9)	<0.001
Openness (mean [SD])	3.2 (0.8)	3.2 (0.9)	3.2 (0.9)	3.2 (0.9)	3.2 (0.8)	0.416
**Adult life alcohol consumption**						
Total weekly units of alcohol (mean [SD])	0 (0)	11.8 (10.7)	13.9 (14.4)	22.4 (28.1)	7.4 (8.1)	<0.001
Weekly units of wine (mean [SD])	0 (0)	7.8 (7.4)	2.7 (3.3)	3.8 (5.5)	2.8 (3.2)	<0.001
Weekly units of beer (mean [SD])	0 (0)	2.8 (3.5)	9.8 (11.6)	5.4 (9.6)	3.0 (3.2)	<0.001
Weekly units of spirits (mean [SD])	0 (0)	1.2 (1.9)	1.4 (3.0)	13.2 (15.3)	1.5 (2.5)	<0.001
Number of years with weekly extreme binge drinking ^b^ (mean [SD])	0 (0)	2.4 (7.7)	5.6 (10.9)	11.5 (14.8)	2.3 (6.4)	<0.001
**Other health behaviors**						
Pack-years of smoking (mean [SD])	16.5 (33.9)	15.5 (24.5)	23.2 (26.0)	43.3 (46.8)	20.8 (33.0)	<0.001
Physical inactivity in leisure time (follow-up) (N [%])	9 (13.2)	79 (6.7)	110 (10.6)	12 (21.4)	13 (12.6)	<0.001
Number of years with weekly use of psychoactive drugs in adult life (mean [SD])	1.1 (5.2)	0.3 (2.8)	1.6 (6.2)	5.4 (10.4)	1.6 (6.6)	<0.001
**Morbidity**						
Alcohol-related hospital diagnosis (N [%])	NA	47 (4.0)	134 (12.9)	20 (35.7)	7 (6.8)	<0.001
Mental disorder hospital diagnosis (N [%])	36 (52.9)	318 (26.8)	414 (39.8)	32 (57.1)	41 (39.8)	<0.001
Charlson Comorbidity Index score ^c^ (mean [SD])	1.1 (1.7)	0.7 (1.6)	1.0 (1.6)	1.3 (1.4)	1.3 (2.4)	<0.001

One unit of alcohol = 12 g of pure alcohol. IQ = intelligence quotient calculated using Børge Priens Prøve test scores. NA: too few individuals in the cell to allow for showing the specific number. ^a^ Chi-squared test for categorical variables and one-way ANOVA for continuous variables. ^b^ Defined as consuming 10 units of alcohol or more on the same occasion. ^c^ Calculated based on information from the Danish National Patient Registry.

**Table 2 nutrients-16-03714-t002:** Associations between adult-life wine, beer, and spirits consumption and IQ changes in men (N = 2456).

		Unadjusted		Model 1		Model 2	
	N	B (95% CI)	*p*-Value	B (95% CI)	*p*-Value	B (95% CI)	*p*-Value
**Weekly units of wine**							
0 units	382 (15.6)	Ref.	-	Ref.	-	Ref.	-
1–7 units	1595 (64.9)	0.32 (−0.72; 1.37)	0.544	−0.25 (−1.40; 0.90)	0.674	1.02 (−0.04; 2.08)	0.061
8–14 units	346 (14.1)	−0.04 (−1.33; 1.40)	0.957	−0.44 (−1.88; 0.99)	0.546	1.47 (0.14; 2.81)	0.030
≥15 units	133 (5.4)	−2.03 (−3.88; −0.18)	0.031	−2.35 (−4.24; −0.46)	0.015	0.27 (−1.49; 2.03)	0.766
**Weekly units of beer**							
0 units	224 (9.1)	Ref.	-	Ref.	-	Ref.	-
1–7 units	1729 (70.4)	0.32 (−0.98; 1.62)	0.632	0.11 (−1.23; 1.47)	0.864	0.29 (−0.93; 1.52)	0.640
8–14 units	288 (11.7)	−0.16 (−1.79; 1.47)	0.847	−0.31 (−1.97; 1.35)	0.713	−0.76 (−2.29; 0.75)	0.321
≥15 units	215 (8.8)	−1.44 (−3.20; −0.31)	0.106	−1.45 (−3.24; 0.34)	0.112	−1.33 (−2.97; 0.32)	0.114
**Weekly units of spirits**							
0 units	1063 (43.3)	Ref.	-	Ref.	-	Ref.	-
1–7 units	1318 (53.7)	0.49 (−0.27; 1.24)	0.208	0.55 (−0.26; 1.36)	0.185	0.82 (0.08; 1.56)	0.030
8–14 units	48 (2.0)	−0.82 (−3.53; 1.89)	0.552	−0.10 (−2.86; 2.67)	0.946	1.30 (−1.22; 3.83)	0.312
≥15 units	27 (1.1)	−1.58 (−5.16; 1.99)	0.385	−0.36 (−4.01; 3.29)	0.846	−0.98 (−4.31; 2.35)	0.563

IQ = intelligence quotient calculated using Børge Priens Prøve test scores. Model 1: adjusted for consumption of other alcoholic beverages (i.e., mutual adjustment by including weekly units of wine, beer, and spirits in the same model). Model 2: Model 1 + adjusted for year of birth, age at follow-up, retest interval length, young adulthood IQ, years of education, and personality. One unit of alcohol = 12 g of pure alcohol.

## Data Availability

Data are not publicly available. To request access to the data from the Lifestyle and Cognition Follow-up study 2015, a formal application should be sent to the person responsible (Merete Osler: meos@sund.ku.dk).

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
