# Peer review of "Differences in Associations of Three Types of Alcoholic Beverages with Age-Related Cognitive Decline in Men"

_nutrients, 2024, doi:10.3390/nu16213714_

Round 1
Reviewer 1 Report
Comments and Suggestions for Authors
This study aims to study the relationship between lifetime alcohol consumption and age-associated cognitive decline, taking into account the preferred type of beverage.
The authors use secondary data, which entails some limitations in terms of the precision of the information collected and its suitability for the objective to be achieved. In this case, furthermore, the data are limited to a single sex, which limits the generalization of results from the start.
The main problem, however, is the rationale of the study.
Introduction
The introduction mentions only two studies relevant to the state of the art, so a broader review of the literature is lacking, perhaps with reference to a meta-analysis or updated systematic review on the subject.
There is insufficient justification as to why the preferred type of beverage may be related to cognitive decline (e.g., neurotoxicity, association with lifestyles, ...). Nor is the relevance of the results to the general population noted.
Method
In relation to the sample, the selection procedure entails important biases. Firstly, the original sample was already biased by the requirements for military service; secondly, only 13% of the sample agreed to collaborate (it is not indicated what characteristics they had, nor the reasons for non-participation); thirdly, as can be seen in Table 1, the groups created in relation to the type of consumption are very different from each other. Of the 25 variables listed in the table, there are significant differences in 17 of them (68%).
Regarding the measurement instruments, the authors do not provide information on the psychometric properties or characteristics of the questionnaire used to evaluate consumption (they refer to another publication), although it is essential to assess the results, as it is data collected retrospectively.
Regarding the measurement of cognitive decline, the authors use an instrument designed for another purpose (estimation of IQ) and used for another purpose, under conditions that are not detailed (which professionals administered it, under what conditions...) . This is a relevant issue because the deterioration associated with age does not affect all cognitive functions equally, so it is relevant to know exactly which functions are evaluated.
Statistical analysis
The authors have made considerable effort to explain the categorization of consumption variables, but have not justified the adjustment variables. Especially striking is the case of personality. Neither the introduction nor the discussion explains the relevance that this variable may have in relation to the cognitive decline associated with age. However, other variables (which differ between groups) such as physical activity are not considered relevant. The authors should explain the rationale of the adjustment variables considered.
Discussion
Regarding the discussion, it would be interesting if the authors offered a more extensive explanation of the underlying mechanisms behind the associations between alcohol consumption and age-related cognitive decline, given that this was the objective of the study.
The authors point out some of the limitations of the work, but the question is, given these limitations, does the study make sense? Are the results generalizable? What implications do they have for public health? Based on these results, would the authors recommend moderate alcohol consumption to prevent age-related cognitive decline? Or is it a variable associated with a certain lifestyle and is it this style, and not specifically alcohol consumption, that explains the results? Do any of the other variables (e.g., lifetime physical activity) have a greater protective effect?
Author Response
Author's Reply to the Review Report
Comment: This study aims to study the relationship between lifetime alcohol consumption and age-associated cognitive decline, taking into account the preferred type of beverage.
Response: Thank you for your comments on the paper. We have replied to them below.
Comment: The authors use secondary data, which entails some limitations in terms of the precision of the information collected and its suitability for the objective to be achieved. In this case, furthermore, the data are limited to a single sex, which limits the generalization of results from the start.
Response: Thank you for the comment. It appears that we have not been sufficiently clear in the description of the data. The LiKO-15 study was designed with the specific aim to collect detailed information on alcohol consumption throughout adult life to enable investigation of the influence of alcohol and alcohol-related disorders on cognitive aging. Thus, this study does not involve secondary data. The overall aim of the LiKO-15 study has been included in the revised version of the paper to outline this fact: “The overall aim of the LiKO-15 study was to investigate alcohol and cognition, and a large proportion of men with psychiatric hospital diagnoses, including alcohol use disorders, were invited to participate.” Moreover, we agree that it is a limitation of the study that only mere were included, which is also described in the discussion section.
Comment: The main problem, however, is the rationale of the study.
Response: We are not entirely sure what the reviewer specifically refers to regarding problems with the rationale of the study, but we have addressed all the specific questions below.
Comment: Introduction
The introduction mentions only two studies relevant to the state of the art, so a broader review of the literature is lacking, perhaps with reference to a meta-analysis or updated systematic review on the subject.
Response: We agree that is important to mention all the studies relevant to the specific paper and preferably meta-analyses or systematic reviews. Nevertheless, the literature on alcohol consumption and age-related cognitive decline (and not just cognition) is sparse and when it comes to specific types of alcoholic beverages and age-related cognitive decline the literature is even more limited. We have identified a meta-analysis of the association between wine consumption and cognitive decline (Lucerón-Lucas-Torres et al., 2022). However, most studies included in this analysis focus on the association between wine consumption and dementia in contrast to studies of non-pathological age-related cognitive changes, and no attempt was made to evaluate cognitive changes quantitatively. In fact, this was not possible in most studies because cognition was not assessed in early adult life. Early assessment is crucial because it is well-documented that young adult individual differences in cognitive ability are stable across most of the lifespan and it is known that at least in some countries wine drinking is associated with higher cognitive ability (Mortensen et al., 2001). Thus, cognitive differences in old age between wine and beer drinkers may reflect lifelong differences in cognitive ability and not necessarily effects of the types of alcohol on cognitive ageing. To make a clear interpretation, you either need an early cognitive baseline assessment or repeated assessment in old age to study decline directly, and the studies we have included in the introduction are among the few studies that have focused on evaluating quantitative cognitive changes.
References:
Lucerón-Lucas-Torres, M., Cavero-Redondo, I., Martínez-Vizcaíno, V., Saz-Lara, A., Pascual-Morena, C., & Álvarez-Bueno, C. (2022). Association Between Wine Consumption and Cognitive Decline in Older People: A Systematic Review and Meta-Analysis of Longitudinal Studies. Frontiers in Nutrition, 9, 863059. https://doi.org/10.3389/fnut.2022.863059
Mortensen, E. L., Jensen, H. H., Sanders, S. A., & Reinisch, J. M. (2001). Better psychological functioning and higher social status may largely explain the apparent health benefits of wine: A study of wine and beer drinking in young Danish adults. Archives of Internal Medicine, 161(15), 1844–1848. https://doi.org/10.1001/archinte.161.15.1844
Comment: There is insufficient justification as to why the preferred type of beverage may be related to cognitive decline (e.g., neurotoxicity, association with lifestyles, ...).
Response: There is a relatively large literature on alcohol and health, in particular the potential health benefits of wine. The recent meta-analysis (Lucerón-Lucas-Torres et al., 2022) also concerns the potential benefits of wine on the risk of dementia or cognitive impairment. Since the analysis concludes that there is potential protective effect of wine but does not evaluate cognitive changes quantitatively and does include studies with early adult cognitive assessment, we find that the current study is well-motivated. We modified the introduction to make this clearer: “The observed associations between alcohol consumption and age-related cognitive decline may be caused by alcohol per se, other compounds in alcoholic beverages or confounding factors such as social status or smoking. If it is alcohol per se that influences age-related cognitive decline, similar associations would be expected across different alcoholic beverages. A recent meta-analysis based on 12 studies has concluded that wine consumption may be associated with lower risk of dementia and cognitive impairment [10]. However, only few studies have investigated whether associations with non-pathological age-related cognitive decline (i.e., including at least two cognitive assessments) vary for different types of alcoholic beverages [2,11].”
Comment: Nor is the relevance of the results to the general population noted.
Response: Since alcohol is widely used in most societies, any differential effects of various types of alcohol on age-related cognitive decline would have important public health implications. We have made this explicit in the introduction: “Determination of the effects of various types of alcohol on cognitive aging is important for public health, as it can inform guidelines and risk assessments, ultimately contributing to better prevention strategies and healthier aging.”
Comment: Method
In relation to the sample, the selection procedure entails important biases. Firstly, the original sample was already biased by the requirements for military service; secondly, only 13% of the sample agreed to collaborate (it is not indicated what characteristics they had, nor the reasons for non-participation);
Response: We agree that the selection procedure entails important biases and in the discussion section of the paper, we have also mentioned selection of participants as one of the main limitations of the study. In the discussion section, we also refer to a previous study where we used LiKO-15 data, in which we found that participants constituted a selected sample on several parameters: “Participation analyses—published in a previous paper [3]—revealed that this self-selection into the study has led to a study sample with higher school educational level, higher baseline intelligence test scores, lower prevalence of mental disorders and other morbidities than those invited to participate.” We do not have information on the reasons why so many of the invited individuals chose not to participate, but given that the examination was approximately two hours long and required that the men showed up at a specific location for cognitive testing and completion of a lengthy questionnaire, it may not be surprising that some men did not participate. Information on this has been included in the revised version of the paper: “Unfortunately, we lack information on the reasons for non-participation, but considering the approximately two-hour duration of the examination, the requirement to travel to a specific location for cognitive testing, and the lengthy questionnaire involved, it is plausible that these factors contributed to the rather low participation rate”.
Comment: …thirdly, as can be seen in Table 1, the groups created in relation to the type of consumption are very different from each other. Of the 25 variables listed in the table, there are significant differences in 17 of them (68%).
Response: We agree that Table 1 illustrates that men with different alcoholic beverage preferences differ in many respects. This is also underlined in the conclusion of the study stating that ‘These results should be interpreted with caution… because differences among alcoholic beverages in the associations with cognitive decline may not only reflect effects of specific compounds, but also inherent differences on both known and unknown characteristics between men who primarily consume wine, beer, or spirits.’
Comment: Regarding the measurement instruments, the authors do not provide information on the psychometric properties or characteristics of the questionnaire used to evaluate consumption (they refer to another publication), although it is essential to assess the results, as it is data collected retrospectively.
Response: The questions on alcohol consumption do not form a psychometric scale, but there is an issue concerning the precision or reliability of the information obtained when self-report information is collected for most of the adult life span. We have not investigated this issue, but the one available study based on a Whitehall II sample found the retest reliability of this type of information to be relatively high (Bell & Britton, 2015). As described in the paper, our study is only based on the assumption that the obtained information is sufficient to categorize participants into consumption groups (thus, the exact numbers reported by each participant is not critical). Furthermore, the British study of reliability included teenagers, while we only included the adult life span from age 26. In adult life, alcohol preferences and consumption can be assumed to be relatively stable, and consequently relatively easy to remember and report retrospectively. In conclusion, we consider the detailed information on adult-life alcohol consumption a significant strength when our study is compared to the many studies that only have information on current alcohol consumption.
Reference: Bell S, Britton A. Reliability of a retrospective decade-based life-course alcohol consumption questionnaire administered in later life. Addiction 2015;110:1563–73. https://doi.org/10.1111/add.13012.
Comment: Regarding the measurement of cognitive decline, the authors use an instrument designed for another purpose (estimation of IQ) and used for another purpose, under conditions that are not detailed (which professionals administered it, under what conditions...) . This is a relevant issue because the deterioration associated with age does not affect all cognitive functions equally, so it is relevant to know exactly which functions are evaluated.
Response: The BPP has been used by the Danish military for more than 50 years, and for almost as long, research based on this test has been conducted and published in international papers. In the paper, we mention that the total BPP score has been found to correlate 0.82 with the WAIS intelligence test, which is the most widely used clinical test of intelligence. This corroborates the validity of the BPP as a measure of intelligence, but the BPP only consists of 4 subtest that all measure aspects of fluid intelligence. In some contexts, this may be a disadvantage, but it is a strength in studies of cognitive ageing because age-related changes in cognition primarily affect various aspects of fluid intelligence. The BPP is a group test administered by special trained military staff at the draft board, and such military staff also administered the test at the midlife follow-up. To accommodate the reviewer’s concern, we have expanded the description of the test in the method section of the manuscript: “Changes in IQ scores from young adulthood to late midlife were used to assess age-related cognitive decline. The intelligence test that participants completed at the draft board examination and at the follow-up examination was Børge Priens Prøve (BPP), which is a 45 minute global test of fluid intelligence. The test was administered by specially trained military staff at the draft board examination, and such staff also administered the test at the midlife follow-up. The test consists of a total of 78 items within subtests of letter matrices, verbal analogies, number series, and geometric figures [12]. The total BPP score has shown to be highly correlated (r=0.82) with the score on the Wechsler Adult Intelligence Scale (WAIS) [15]. The total score on BPP ranges from 0 to 78 points, and both sets of raw test scores (from young adulthood and late midlife, respectively) were linearly standardized to an IQ scale with a study sample mean of 100 and standard deviation of 15 in young adulthood. The IQ scale is generally known and frequently used; hence, making our results easier to interpret and compare to results in other studies.” In conclusion, the BPP is a comprehensive cognitive test that can effectively be used to compare cognitive ability in young adulthood and late midlife. In contrast, commonly used screening instruments like the MMSE tend to show less accurate performance in young adults and in fact also in many older individuals.
Comment: Statistical analysis
The authors have made considerable effort to explain the categorization of consumption variables, but have not justified the adjustment variables. Especially striking is the case of personality. Neither the introduction nor the discussion explains the relevance that this variable may have in relation to the cognitive decline associated with age. However, other variables (which differ between groups) such as physical activity are not considered relevant. The authors should explain the rationale of the adjustment variables considered.
Response: It is important to note that the information on health-related behaviors, including physical activity, was only available for the midlife follow-up, and since lifestyle is likely to change from young adulthood to late midlife, it is problematic to adjust for physical activity or other lifestyle factors that may be influenced by alcohol consumption during adult life. We have noted this in the ‘Additional variable’ section “Additional variables on alcohol consumption pattern, other health behaviors and morbidity were also included; however, these variables were not included in the primary analyses because they are not necessarily confounders but could also mediate the associations”. Moreover, we have included adjustment for health behaviors, including physical activity, in the supplementary table, but the overall conclusions based on these results are the unchanged. There is a relatively large literature on personality and physical health and a number of studies on personality and age-related cognitive decline, especially on neuroticism, stress and cognitive decline. Personality traits are assumed to be stable during the adult life, and consequently it may be less risky to adjust for personality traits, which may influence alcohol consumption in adulthood. We have included the following description of this in the revised version of the paper: “Previous studies have shown that personality may influence both alcohol consumption and cognitive aging. Although personality was only assessed at the follow-up examination, it was included as a covariate because personality traits are assumed to be relatively stable during adult life.”
Comment: Discussion
Regarding the discussion, it would be interesting if the authors offered a more extensive explanation of the underlying mechanisms behind the associations between alcohol consumption and age-related cognitive decline, given that this was the objective of the study.
Response: Several pathways have been suggested to explain associations between heavy alcohol consumption and age-related cognitive decline (cardiovascular disease, direct neurotoxic effects on the brain, and increased inflammatory response). However, this is not the focus of our paper, which aims to identify any positive or negative effects of the three types of alcohol. The paper contains the following paragraph on mechanisms, which we believe is sufficient at the present state of knowledge: “Dissimilar associations with cognitive decline observed across different alcoholic beverages could indicate that it is not alcohol per se that influences cognitive decline. Thus, compounds other than ethanol in wine and spirits could explain the apparently beneficial effects of moderate consumption on cognitive decline. Primarily in relation to wine, there has been a focus on explaining the apparently beneficial effects on health, where polyphenols (besides ethanol) seem to be central in explaining red wine’s antioxidant, anti-inflammatory, and cytoprotective properties [10, 22]. However, in our study, significant associations with less cognitive decline were only observed for wine and spirits when adjusted for potential confounders, and direct statistical test of differences between corresponding regression coefficients for the three types of beverages were only significant for moderate consumption of wine and beer. Furthermore, this test was only significant at the 5% percent level, and this was in fact the case for all significant results in table 2. Thus, none of these tests would remain significant if adjustment was made for the relatively large number of statistical tests (e.g., Bonferroni correction)”.
Comment: The authors point out some of the limitations of the work, but the question is, given these limitations, does the study make sense? Are the results generalizable? What implications do they have for public health? Based on these results, would the authors recommend moderate alcohol consumption to prevent age-related cognitive decline? Or is it a variable associated with a certain lifestyle and is it this style, and not specifically alcohol consumption, that explains the results? Do any of the other variables (e.g., lifetime physical activity) have a greater protective effect?
Response: As the reviewer points out, we describe a number of limitations of the study. In spite of these limitations, we decided to conduct the study. An important reason is that we believe that there are few databases with sufficiently detailed information on alcohol and age-related cognitive decline to analyze effects of various types of alcohol and with fewer limitations than our study. Given the limitations of our study and the almost non-existent literature on the topic, we find it premature to draw public health conclusions.
Reviewer 2 Report
Comments and Suggestions for Authors
This study made by Grønkjær and colleagues on alcohol consumption and age-related cognitive decline in men offers valuable insights.
The study relies on self-reported data about alcohol consumption, which introduces potential recall bias. Participants may not accurately remember or report their alcohol intake over long periods, particularly as the data are retrospective and span from age 26 to late midlife.
The presented study sample includes only Danish men, which limits the generalizability of the findings to other populations, such as women, other age groups, or individuals from different cultural backgrounds with different drinking patterns and genetic predispositions.
Although the study adjusts for several important confounders (education, year of birth, retest interval, intelligence in young adulthood and personality), there may be residual confounding. The observed associations might be influenced by other unmeasured factors like socioeconomic status, overall diet, lifestyle factors (e.g., physical activity, smoking), or genetic predisposition to both alcohol consumption and cognitive decline.
The authors acknowledge the potential for weak statistical power in some comparisons, particularly for spirits, which could lead to type II errors (failing to detect a true effect). The weak power makes it difficult to confidently conclude that the observed associations are not due to chance.
The study design is observational, meaning it cannot establish causality. The associations between alcohol consumption and cognitive decline do not prove that alcohol consumption directly influences cognitive function. Other underlying factors could be responsible for both alcohol consumption habits and cognitive outcomes.
The differences in cognitive decline associated with different types of alcoholic beverages (wine, beer, and spirits) may not solely reflect the effects of alcohol or specific compounds (e.g., polyphenols in wine). They could also reflect lifestyle differences among consumers of different beverages. For example, wine drinkers may have different dietary or social habits than beer drinkers, which may contribute to cognitive health.
The paper identifies moderate wine and spirits consumption as being potentially protective, but the definition of "moderate" varies culturally, and the threshold of 8-14 units per week for wine may not apply universally. Additionally, health guidelines regarding safe alcohol consumption differ between countries, which complicates the interpretation and application of the findings.
The authors suggest that certain types of alcohol (wine, spirits) may be associated with less cognitive decline, but it does not provide mechanistic insights into why this might be the case. More detailed biological or biochemical studies would be necessary to understand the specific compounds or pathways involved.
I think that the title should be completed with more information.
Author Response
Author's Reply to the Review Report
Comment: This study made by Grønkjær and colleagues on alcohol consumption and age-related cognitive decline in men offers valuable insights.
The study relies on self-reported data about alcohol consumption, which introduces potential recall bias. Participants may not accurately remember or report their alcohol intake over long periods, particularly as the data are retrospective and span from age 26 to late midlife.
Response: We refer to the paragraph on this problem in the strengths and limitations section of the paper. Furthermore, reviewer 1 also raised this issue, and we have copied our answer: The questions on alcohol consumption do not form a psychometric scale, but there is an issue concerning the precision or reliability of the information obtained when self-report information is collected for most of the adult life span. We have not investigated this issue, but the one available study based on a Whitehall II sample found the retest reliability of this type of information to be relatively high (Bell & Britton, 2015). As described in the paper our study is only based on the assumption that the obtained information is sufficient to categorize participants into consumption groups (thus, the exact numbers reported by each participant is not critical). Furthermore, the British study of reliability included teenagers, while we only included the adult life span from age 26. In adult life, alcohol preferences and consumption can be assumed to be relatively stable, and consequently relatively easy to remember and report retrospectively. In conclusion, we consider the detailed information on adult-life alcohol consumption a significant strength when our study is compared to the many studies that only have information on current alcohol consumption.
Reference: Bell S, Britton A. Reliability of a retrospective decade-based life-course alcohol consumption questionnaire administered in later life. Addiction 2015;110:1563–73. https://doi.org/10.1111/add.13012.
Comment: The presented study sample includes only Danish men, which limits the generalizability of the findings to other populations, such as women, other age groups, or individuals from different cultural backgrounds with different drinking patterns and genetic predispositions.
Response: We fully agree with the reviewer about the potential limitations of generalizability to women and individuals with different cultural background and drinking patterns. To accommodate the reviewer’s concerns, we have added the following sentence to the ‘strength and limitations’ section: “However, our results may not be generalized to women or countries with different drinking cultures.”
Comment: Although the study adjusts for several important confounders (education, year of birth, retest interval, intelligence in young adulthood and personality), there may be residual confounding. The observed associations might be influenced by other unmeasured factors like socioeconomic status, overall diet, lifestyle factors (e.g., physical activity, smoking), or genetic predisposition to both alcohol consumption and cognitive decline.
Response: There is much evidence that education is closer related to cognition than broader measures of social status, and consequently, we do not believe that additional control of social status would change the results considerably. A good measure of diet would have been informative and potentially important, but such measures are complex – in particular a measure of diet through the adult lifespan. As mentioned in our response to reviewer 1, table S1 includes a model controlling for health behaviors such as pack-years of smoking, physical inactivity in leisure time, and number of years with weekly use of psychoactive drugs. It is a supplementary analysis since these variables are not necessarily confounders but could also mediate the associations, and for physical activity, information was only available at follow-up and consequently may have been influenced by alcohol drinking patterns through adult life and not vice versa. Finally, we agree that genetic predisposition may play a role – but to our knowledge it is not known whether certain genetic dispositions are associated with both alcohol preferences and age-related cognitive decline.
Comment: The authors acknowledge the potential for weak statistical power in some comparisons, particularly for spirits, which could lead to type II errors (failing to detect a true effect). The weak power makes it difficult to confidently conclude that the observed associations are not due to chance.
Response: As the reviewer points out, we acknowledge that statistical power problems may have influenced the results, which is also made clear in the following paragraph in the discussion section: “Finally, some of our results may be related to subgroup sample sizes and associated statistical power. For example, the 15 units or more subgroups included 133 men for wine and 215 for beer, but only 27 men for spirits. Thus, apparent differences in the effects of the three types of beverages on cognitive decline may to some extent reflect statistical power. The estimated coefficients in some cases corroborate this interpretation and also suggest that even significant effects of alcohol consumption are small compared to the theoretical IQ standard deviation of 15 (adjusted for covariates the largest effect was 1.47 associated with 8–14 units of wine, corresponding to 0.10 standard deviation). These positive effects are smaller than the negative effects on cognitive decline of more than 28 units a week observed in the same sample in a study of total alcohol consumption [26], and a particular weakness of the current study is that we were unable to analyze effects of higher levels of consumption in detail. Thus, we cannot preclude that differences in the effects of the three beverage types could be demonstrated for higher consumption levels. “
Comment: The study design is observational, meaning it cannot establish causality. The associations between alcohol consumption and cognitive decline do not prove that alcohol consumption directly influences cognitive function. Other underlying factors could be responsible for both alcohol consumption habits and cognitive outcomes.
Response: We agree that this is a basic limitation of observational studies – in particular in an area where there are very few comparable studies.
Comment: The differences in cognitive decline associated with different types of alcoholic beverages (wine, beer, and spirits) may not solely reflect the effects of alcohol or specific compounds (e.g., polyphenols in wine). They could also reflect lifestyle differences among consumers of different beverages. For example, wine drinkers may have different dietary or social habits than beer drinkers, which may contribute to cognitive health.
Response: We fully agree and indeed this is made clear in the very last sentences of the paper: “These results should be interpreted with caution because of weak statistical power for some comparisons and because differences among alcoholic beverages in the associations with cognitive decline may not only reflect effects of specific compounds, but also inherent differences on both known and unknown characteristics between men who primarily consume wine, beer, or spirits”
Comment: The paper identifies moderate wine and spirits consumption as being potentially protective, but the definition of "moderate" varies culturally, and the threshold of 8-14 units per week for wine may not apply universally. Additionally, health guidelines regarding safe alcohol consumption differ between countries, which complicates the interpretation and application of the findings.
Response: In principle, the reviewer is right, but the presentation in table 2 is not dependent on interpretation of “moderate drinking”, since number of units is used as basis for categorizing consumption of each type of alcohol in four groups. Thus, the table can be interpreted from any perspective on what moderate drinking include. In the text, we used the term moderate drinking, but the term is defined as 1-7 units in the results section.
Comment: The authors suggest that certain types of alcohol (wine, spirits) may be associated with less cognitive decline, but it does not provide mechanistic insights into why this might be the case. More detailed biological or biochemical studies would be necessary to understand the specific compounds or pathways involved.
Response: This is true, and it underlines the fact that much more research is needed in this area.
Comment: I think that the title should be completed with more information.
Response: The reviewer does not state what information is missing, but to make the title even more precise we have expanded the title to “Differences in associations of three types of alcoholic beverages with age-related cognitive decline in men”
Round 2
Reviewer 1 Report
Comments and Suggestions for Authors
The comments are in the attached file.

Author Response
The author's reply to the review report is in the attached file.

Reviewer 2 Report
Comments and Suggestions for Authors
Considering the major improvements made by the authors in the manuscript and also by the answers given to the comments, I believe it can be published.
Author Response
Comment: Considering the major improvements made by the authors in the manuscript and also by the answers given to the comments, I believe it can be published.
Response: Thank you very much. We are pleased to hear that you also find that the revisions have improved the manuscript. We appreciate your time and effort in reviewing our work.
Round 3
Reviewer 1 Report
Comments and Suggestions for Authors
I have read with interest the authors' responses to my considerations about the rationale for the study and the limitations to internal and external validity. After reviewing their responses and the modifications made, I believe that the limitations are reflected in the manuscript. But, my main concern regarding the study remains. The objective is "to help elucidate the underlying mechanisms behind the associations between alcohol consumption and non-pathological age-related cognitive decline in men ["by examining the influence of different types of alcoholic beverages", indicates the way in which they want to answer the question, it is not the objective]. However, the authors do not justify why different types of beverages should be associated with different effects on cognitive aging (principle of biological plausibility). In the introduction, it is stated that: "The observed associations between alcohol consumption and age-related cognitive decline may be caused by alcohol per se, other compounds in alcoholic beverages or confounding factors such as social status or smoking." Is it possible to elucidate which is the most probable explanation with the results obtained? In the case of wine, an explanation is proposed, but in the case of spirits, no. Similar results, but different mechanisms? How to conclude that “moderate consumption of wine and spirits can mitigate cognitive decline, unlike beer”, without suggesting any underlying mechanism? Why is the effect attributed to beverages and not to other uncontrolled confounding variables (i.e. socioeconomic status)? In summary, if the current wording of the study objective is maintained and the data are analyzed in terms of the effect (causality) of different beverages on age-related cognitive decline, I consider it necessary to justify the principle of biological plausibility. In the attached file you will find the detailed comments.

Author Response

(The authors gave the same response as above.)

Round 4
Reviewer 1 Report
Comments and Suggestions for Authors.